# From Inflammation to Oncogenesis: Tracing Serum DCLK1 and miRNA Signatures in Chronic Liver Diseases

**DOI:** 10.3390/ijms25126481

**Published:** 2024-06-12

**Authors:** Landon L. Moore, Dongfeng Qu, Sripathi Sureban, Stephanie Mitchell, Kamille Pitts, Nasya Cooper, Javid Fazili, Richard Harty, Abdul Oseini, Kai Ding, Michael Bronze, Courtney W. Houchen

**Affiliations:** 1Department of Medicine, University of Oklahoma Health Sciences Center, Oklahoma City, OK 73104, USA; landon-moore@ouhsc.edu (L.L.M.); dongfeng-qu@ouhsc.edu (D.Q.); mssripathi@gmail.com (S.S.); stephanie-mitchell@ouhsc.edu (S.M.); kamille-pitts@ouhsc.edu (K.P.); javid-fazili@ouhsc.edu (J.F.); richard-harty@ouhsc.edu (R.H.); abdul-oseini@ouhsc.edu (A.O.); michael-bronze@ouhsc.edu (M.B.); 2Department of Veterans Affairs Medical Center, Oklahoma City, OK 73104, USA; 3Department of Natural Sciences, Langston University, Langston, OK 73050, USA; nasya.cooper@langston.edu; 4Department of Biostatistics and Epidemiology, University of Oklahoma Health Sciences Center, Oklahoma City, OK 73104, USA; kai-ding@ouhsc.edu

**Keywords:** liver fibrosis, cirrhosis, hepatocellular carcinoma, microRNA, DCLK1

## Abstract

Chronic liver diseases, fibrosis, cirrhosis, and HCC are often a consequence of persistent inflammation. However, the transition mechanisms from a normal liver to fibrosis, then cirrhosis, and further to HCC are not well understood. This study focused on the role of the tumor stem cell protein doublecortin-like kinase 1 (DCLK1) in the modulation of molecular factors in fibrosis, cirrhosis, or HCC. Serum samples from patients with hepatic fibrosis, cirrhosis, and HCC were analyzed via ELISA or NextGen sequencing and were compared with control samples. Differentially expressed (DE) microRNAs (miRNA) identified from these patient sera were correlated with DCLK1 expression. We observed elevated serum DCLK1 levels in fibrosis, cirrhosis, and HCC patients; however, TGF-β levels were only elevated in fibrosis and cirrhosis. While DE miRNAs were identified for all three disease states, *miR-12136* was elevated in fibrosis but was significantly increased further in cirrhosis. Additionally, *miR-1246* and *miR-184* were upregulated when DCLK1 was high, while *miR-206* was downregulated. This work distinguishes DCLK1 and miRNAs’ potential role in different axes promoting inflammation to tumor progression and may serve to identify biomarkers for tracking the progression from pre-neoplastic states to HCC in chronic liver disease patients as well as provide targets for treatment.

## 1. Introduction

Chronic liver disease, with the potential progression to fibrosis, cirrhosis, and eventually hepatocellular carcinoma (HCC), is a significant global health issue due to its high morbidity and mortality [1]. HCC is the fifth leading cause of cancer-related death in the U.S. and the fourth worldwide. Key risk factors for the development of HCC include chronic viral hepatitis (mainly hepatitis C in the US), non-alcoholic steatohepatitis (NASH), and other causes of cirrhosis [2]. Despite improvements in the treatment of hepatitis C, the incidence of cirrhosis and HCC has increased, likely due to the increasing prevalence of NASH and, lately, alcohol-associated liver disease. Interestingly, the incidence of HCC is five times higher among veterans than the general population, partly due to the over-representation of males in this population [3].

The precise molecular mechanisms that govern the progression from normal liver tissue to fibrosis and then to cirrhosis, which is the most prominent precursor to HCC, remains unclear. Unfortunately, the most clinically useful serum marker of HCC, alpha-fetoprotein (AFP), has modest diagnostic accuracy as increased AFP levels could also suggest increased severity of hepatic destruction and subsequent regeneration [4,5]. In particular, the AFP-L3 isoform has been spotlighted due to being elevated in HCC [6]. Another protein of interest is the angiotensin-converting enzyme 2 (ACE2), which has been shown to impact HCC and is modulated by β-catenin activity [7,8]. In these studies, ACE2 protein levels are reduced in HCC patients’ livers and increased expression might improve survival. However, a mechanistic understanding remains elusive. Identifying key molecular factors and pathways in the transition from pre-neoplastic liver fibrosis to HCC may lead to potential diagnostic biomarkers and therapeutic agents for preventing, treating, or reversing chronic liver disease complications [9].

MicroRNAs (miRNAs), crucial regulators, are likely involved in the progression of liver disease. OncomiRs are miRNAs targeting either oncogenes or tumor suppressors and thus have a major impact on tumorigenesis. In HCC, *miR-21* is often upregulated and acts as an oncomiR by regulating tumor growth and invasion, while *miR-122* is typically downregulated and effectively functions as a tumor suppressor by maintaining lipid metabolism and inhibiting tumorigenesis [10]. Additionally, liver fibrosis progression is mediated by fibrogenic miRNAs, like *miR-199a-5p*, which act via TGF-β [11,12], whereas *miR-29* [13] and *miR-150* [14] exhibit anti-fibrotic properties in the liver. In non-alcoholic fatty liver disease (NAFLD), *miR-122* regulates lipid homeostasis, and its dysregulation leads to steatosis, while *miR-34a* contributes to inflammation and progression to NASH [15]. Collectively, miRNAs are integral to liver disease progression, influencing various pathogenic mechanisms and holding potential as diagnostic and therapeutic targets.

Doublecortin-like kinase 1 (DCLK1), a serine–threonine protein kinase, is a marker of tumor stemness in many solid tumor cancers, including HCC [16]. DCLK1 is regulated by various miRNAs, which can modulate its expression and, consequently, its role in cellular processes such as tumorigenesis and cancer progression. For instance, *miR-144* directly targets DCLK1, leading to the downregulation of DCLK1 expression, thereby inhibiting cancer cell growth and invasion [17]. Similarly, *miR-200a* has been shown to suppress DCLK1, which is associated with reduced stemness and tumorigenic potential in pancreatic cancer cells [18]. Conversely, DCLK1 can also influence the expression of downstream miRNAs, creating a feedback loop that impacts various signaling pathways. For example, DCLK1 knockdown in colorectal cancer cells results in the upregulation of tumor suppressor miRNAs such as *miR-143* and *miR-145*, leading to reduced cell proliferation and migration [19]. Thus, the interplay between DCLK1 and miRNAs underscores a complex regulatory network that is crucial for maintaining cellular homeostasis and modulating cancer-related pathways.

The development of HCC is believed to be driven by specific cells that exhibit stem cell qualities, such as the ability to self-renew and transition between cell types (epithelial-mesenchymal transition or EMT) [20,21]. Further, DCLK1 promotes hepatocyte clonogenicity and oncogenic programming via a non-canonical WNT-β-catenin-dependent mechanism [22]. Notably, data from The Cancer Genome Atlas have shown that the WNT pathway oncogene (*CTNNB1*) constitutes about 30% of the significantly mutated genes (SMGs) in HCC [23]. In addition, in both macrophages and epithelial cells, DCLK1 has been linked with the phosphorylation of IKKB and subsequent release of multiple pro-inflammatory cytokines, including IL-6 [24,25,26]. We previously reported a mechanistic association between HCV infection and stemness in liver-derived cells and observed a marked increase in immunoreactive DCLK1 expression in HCV patients with cirrhosis [27,28]. Moreover, we separately reported that DCLK1 is upregulated in cirrhosis and HCC and suggested that the mechanism may be miRNA-mediated [29]. This led us to investigate the correlation between DCLK1 and stage-specific transformation from liver fibrosis, cirrhosis, and HCC in patients, when compared with normal healthy subjects. While blood-based miRNAs and proteins may serve as biomarkers, they may also be used to identify potential therapeutic targets [30]. Here, we describe differentially expressed miRNAs in liver disease, the expression of some correlated with DCLK1 expression levels, and represent potential therapeutic targets for liver disease.

## 2. Results

### 2.1. DCLK1 Levels Are Elevated in the Blood Serum of Patients with Liver Disease

DCLK1 protein levels in the sera of 270 test subjects were analyzed to determine the possible association between DCLK1 expression and the observed stage of liver disease. Within these samples, DCLK1 protein levels were increased ~2.5-fold in patients with fibrosis (*p* value = 0.0018) or cirrhosis (*p* value = 0.0001), and in patients diagnosed with HCC, DCLK1 the serum level was ~2.0-fold greater (*p* value = 0.0005) when compared with controls with no known liver disease (Figure 1A). However, we did not observe any statistically significant differences in DCLK1 protein levels between patients with the different stages of progressive liver disease. 

Having established that DCLK1 is elevated in sera from liver disease patients, we sought to investigate downstream effectors of elevated DCLK1, as well as other potential markers, in order to distinguish the various stages of liver disease. DCLK1 is an upstream factor regulating transforming growth factor-beta (TGF-β) [31] and, given the well-described role of TGF-β in progressive hepatic epithelial fibrosis and HCC [32], we next evaluated TGF-β protein levels in patient samples described above. We observed a statistically significant increase in TGF-β in patients with fibrosis and cirrhosis compared with normal (*p* values of 0.001 and 0.0028, respectively). However, there were no statistically significant differences in TGF-β levels between patients with fibrosis and cirrhosis (*p* value, 0.4167) (Figure 1B). Interestingly, we did not observe any significant increase in TGF-β levels in HCC patients compared with normal patients (*p* value, 0.9997). Rather, there was a significant reduction in TGF-β between fibrosis/cirrhosis patients compared with HCC (*p* value, <0.0001), suggesting that downregulation of TGF-β has the potential to distinguish between pre-neoplastic (fibrosis, cirrhosis) and the malignant transformation observed in HCC.

To validate our cohort sera samples using protein markers previously identified, we next evaluated a well-known marker for liver disease, AFP-L3 [4] and, as such, we observed increased AFP-L3 in fibrosis (2.8-fold), cirrhosis (2.6-fold), and HCC (12.6-fold) patients compared with normal (Figure 1C, *p* values of 0.0041, 0.0083, and <0.0001, respectively). There were no statistically significant differences between cirrhosis and fibrosis (*p* value = 0.9479). However, there was a statistically significant increase in AFP-L3 in HCC compared with cirrhosis (*p* value = 0.0283). Furthermore, we examined ACE2 [7,8] protein serum levels in patients with liver disease relative to control (Figure 1D). There was not a statistically significant change in ACE2 protein levels in sera from patients with fibrosis (*p* value, 0.1388), or cirrhosis (*p* value, 0.3408) compared with control; however, there was increased expression of ACE2 in some cirrhotic patients, thus increasing variability. Increased serum ACE2 was significant in HCC compared with controls (*p* value <0.0001), and ACE2 sera expression was significantly upregulated between fibrosis or cirrhosis versus HCC (*p* values of <0.005 and <0.04, respectively), suggesting that elevated serum ACE2 expression can distinguish between patients with fibrosis/cirrhosis versus HCC.

### 2.2. Serum from Patients with Fibrosis, Cirrhosis, or HCC Display Unique miRNAs

After our protein analysis failed to differentiate between liver disease states, we next analyzed microRNAs (miRNAs) in plasma and serum to identify distinct miRNAs for normal liver, fibrosis, cirrhosis, and HCC through pairwise comparisons [33,34]. Initially, we analyzed the 15 most abundant differentially expressed (DE) miRNAs in each serum. In fibrosis and cirrhosis, *miR-16-5p* was the most abundant, comprising over 83% of total DE miRNAs, with incomplete overlap in the next 14 DE miRNAs (Figure 2A,B). However, in HCC, *miR-126-3p* dominated (77.7%), with a completely different DE miRNA profile from fibrosis and cirrhosis (Figure 2C). These results indicate distinct DE miRNA profiles in each liver disease, offering potential markers for disease progression.

### 2.3. Differential Expression of miRNAs in Liver Disease Progression

To identify markers and potential therapeutic targets for different liver stages/diseases, we analyzed DE miRNAs in serum from fibrosis, cirrhosis, and HCC patients compared with controls. We found several miRNAs significantly upregulated or downregulated in fibrosis (Figure 3A), with four also regulated in cirrhosis, particularly *miR-12136* (Figure 3B). Direct comparison between fibrosis and cirrhosis revealed the differential expression of *miR-12136*, *miR-1246*, and *miR-1290*, alongside four other distinct miRNAs (Figure 3C), indicating potential markers to differentiate these conditions. However, HCC patient serum miRNAs showed no overlap with fibrosis or cirrhosis (Figure 3D), suggesting distinguishing markers for HCC and aligning with previous protein marker observations.

Given that our results show that unique miRNAs are associated with fibrosis, cirrhosis, and HCC, we next closely analyzed the relationships between serum miRNAs and liver disease stage. Comparing fibrosis, cirrhosis, and HCC against normal liver, we identified 14 significant DE miRNAs in fibrosis, 24 in cirrhosis, and 12 in HCC, with 2 common to cirrhosis and HCC and 1 significant across all three liver stages (Figure 3E, Table 1). Amongst the DE miRNAs common to fibrosis and cirrhosis, we observed that some miRNAs showed major expression level changes potentially representing markers for fibrosis progression to cirrhosis. *miR-12136* was elevated 2.7-fold in fibrosis (*p* value, 0.0010) and increased a further 5.5-fold in cirrhosis when compared with control (*p* value, 6.56 × 10^−19^). This represents a 2.6-Log_2_-fold increase (*p* value, 0.0018) in expression between fibrosis and cirrhosis. In contrast, other miRNAs, such as *miR-1246* and *miR-1290*, were upregulated in both fibrosis and cirrhosis but were not significantly different amid the two stages (*p* values of 0.1708 and 0.6963, respectively).

As liver disease progresses to cirrhosis, the risk of HCC increases [35]. Our study identified 24 miRNAs differently expressed in cirrhosis (Table 2). Among these miRNAs, we discovered that most were downregulated in cirrhosis and HCC, including *miR-1299* and *miR205-5p*, which are known tumor suppressors [36,37]. Of the upregulated DE miRNAs, the most significant was *miR-184* which was overexpressed 4.7-fold. Interestingly, another upregulated miRNA, *miR-206*, was further expressed in HCC relative to cirrhosis (4.8-fold). This miRNA has been linked to HCC, through its supporting role there in the progression from cirrhosis [38]. Furthermore, in HCC patients’ serum, we observed that, like with cirrhosis, most miRNAs were downregulated in HCC, with the two most downregulated (2.5-fold) being *miR-150-5p* and *miR-375-3p* (Table 3). We only observed two miRNAs, *miR-132-5p* and *miR-1537-3p* upregulated in HCC, at 3.6-fold and 3.2-fold, respectively.

### 2.4. Differentially Expressed (DE) miRNAs Show DCLK1-Specific Differences in Liver Disease

Given that DCLK1 is upregulated in fibrosis, cirrhosis, and HCC (Figure 1A), we examined if any previously identified DE miRNAs correlated with DCLK1 serum protein levels. Based on serum DCLK1 levels, patient samples were grouped as either DCLK1 high or DCLK1 low (Appendix A). Principal component analysis on the 500 miRNAs with the highest variance (Figure 4A) showed that fibrosis and cirrhosis clustered together in a manner that was distinct from both normal and HCC miRNAs. Furthermore, we observed similar clustering of DE miRNAs by hierarchal cluster analysis (Figure 4B). This analysis revealed that, in fibrosis, *miR-12136*, *miR-1246*, and *miR-1290* grouped and were elevated, whereas in cirrhotic patients *miR-184* and *miR-206* clustered. Interestingly, the downregulation of miRNAs was more prominent in HCC, as some miRNAs were largely upregulated, other than *miR-512-3p*. To assess DCLK1 involvement more directly in miRNAs’ differential expression we performed pairwise analysis and discovered three miRNAs, *miR-206*, *miR-184*, and *miR-1246*, that were significantly differentially expressed between the DCLK1 high and DCLK1 low samples (Table 4).

## 3. Discussion

The development of liver diseases involves complex interactions between various cell types, signaling pathways, and molecular regulators, such as miRNAs, cytokines, growth factors, and transcription factors. Liver fibrosis involves HSC activation and the influence of fibrogenic cytokines, like TGF-β, with *miR-21* and *miR-199a-5p* promoting fibrosis and *miR-29* and *miR-150* exhibiting anti-fibrotic properties [12,13,14]. In NAFLD and NASH, insulin resistance and inflammatory cytokines, such as TNF-α and IL-6, are key drivers, with *miR-122* and *miR-34a* playing significant roles in lipid metabolism and inflammation [15]. In liver cirrhosis, chronic inflammation and fibrosis lead to architectural distortion, with circulating miRNAs like *miR-29* and *miR-199a-5p* serving as biomarkers for liver damage [39]. In HCC, oncogenes like c-MYC and RAS, along with tumor suppressors such as p53, play crucial roles, while miRNAs acting as OncomiRs modulate these genes or directly promote tumorigenesis [40,41]. These regulatory relationships underscore the multifaceted nature of liver disease progression.

Fibrosis and cirrhosis are pre-neoplastic phenotypes and many patients with cirrhosis may already have developed HCC at diagnosis [42,43]. The clinical accuracy of AFP-L3 as a biomarker for HCC is modest (sensitivity 39–65% and specificity 76–94%) [44], with nearly one third of early-stage HCC cases being missed when AFP is used alone [45]. Further, serum AFP levels can be elevated in nonmalignant liver diseases, such as acute hepatitis [46]. AFP-L3’s specificity for malignant hepatocytes makes it a possible biomarker for distinguishing HCC from cirrhosis, though its sensitivity alone is insufficient for comprehensive screening [47]. Therefore, while AFP-L3 alone has diagnostic value, it is more effective when used with other biomarkers to enhance accuracy. Thus, there is an urgent need to identify early biomarkers for detecting patients before the development of HCC.

We previously reported a link between DCLK1, HCV-induced inflammation, and stemness in liver-derived clonogenic cells [27,28]. DCLK1 has been reported to influence tumor stemness, EMT, and metastasis in several solid tumors [17,29,48,49,50,51,52], and has been linked to pro-inflammatory NF-κB signaling activation by interacting with IKKβ [31]. These findings strongly suggest a potentially critical role for DCLK1 in hepatocyte response to inflammation, promoting hepatic tumor stemness. Given DCLK1’s involvement in inflammation-related cancer initiation and development, we investigated its role as a potential biomarker and therapeutic target in the progression from normal hepatocytes to HCC [53]. We hypothesized that DCLK1 contributes to HCC progression following chronic inflammatory hepatic injury. Our study found increased serum DCLK1 levels in patients with fibrosis, cirrhosis, and HCC compared with controls. However, DCLK1 protein levels did not show stage-specific differences to distinguish between these clinical phenotypes. This indicates that, while DCLK1 serum levels are elevated in chronic liver disease, they do not differentiate between progressive stages of liver disease. The consistent DCLK1 expression across different stages of liver disease may be due to its stable roles in stem cell regulation, inflammation, and tumorigenesis. This stability suggests that DCLK1 levels do not significantly fluctuate with disease stages, highlighting its potential as a steady biomarker or therapeutic target [47].

In contrast with increased DCLK1 in all three conditions, we found that its downstream effector, TGF-β, was only upregulated in fibrosis and cirrhosis [31]. This suggests that the DCLK1/TGF-β pathway is primarily active in the preneoplastic phase of chronic liver disease. The reduction of TGF-β during the transition from cirrhosis to HCC may reflect its dual role as both an oncogene and a tumor suppressor [54]. The diagnostic efficacy of TGF-β in differentiating HCC from pre-neoplastic stages is due to its link to hepatocyte destruction and activation of hepatic stellate cells, critical in the transition to HCC [55]. Conversely, the HCC biomarker AFP-L3 [5] and the fibrosis marker ACE2 [56] showed significantly elevated serum levels in HCC compared with fibrosis or cirrhosis. Thus, fibrosis and cirrhosis are best identified by an elevated DCLK1/TGF-β axis, while HCC is better distinguished via AFP-L3 and ACE2. These results suggest that, while DCLK1 is essential in all stages of chronic liver disease, its role in HCC progression involves a pathway distinct from TGF-β.

Disease progression alters miRNA profiles as liver cells respond to injury, fibrosis, and cancerous changes. miRNA expression is tissue-specific, reflecting the states of hepatocytes, stellate cells, and immune cells. Inflammation and immune responses also modulate miRNA levels, with specific miRNAs regulating these processes [47,55]. We aimed to identify the unique DCLK1-specific downstream signaling pathways that indicate differences from cirrhosis to HCC. miRNAs, due to their functional roles, stability, and small size, are excellent candidates for disease markers and are detectable in blood, plasma, serum, saliva, and urine [33,57,58,59,60]. Recent reports show that elevated miRNAs correlate with liver cirrhosis, while others reflect liver inflammation or damage [61,62,63]. This suggests specific miRNA signatures may distinguish and define disease progression to fibrosis [57,64]. We hypothesized that specific miRNAs regulated by DCLK1 would serve as key factors in the progression from fibrosis to cirrhosis and HCC.

In this study, we identified several serum miRNAs that are correlated with disease progression in chronic liver disease. These miRNAs exhibit differential expression levels, allowing for future functional and mechanistic evaluation. For instance, *miR-16-5* is abundant in both fibrosis and cirrhosis, consistent with its role in resolving fibrosis [65], but its expression decreased with progression to HCC, where it acts as a tumor suppressor [66,67]. Similarly, *miR-1246* is differentially expressed in fibrosis and cirrhosis but is absent in HCC. Notably, the miRNA profile changed between fibrosis and cirrhosis, indicating a shift in molecular factors. For example, *miR-12136* was significantly upregulated in fibrotic and cirrhotic patients when compared with normal subjects, showing dramatic increases in cirrhotic patients compared with fibrosis alone. This suggests a role for *miR-12136* in the early detection of fibrosis and when monitoring progression to cirrhosis. Further studies are needed to understand the mechanism of *miR-12136* in liver disease progression. We found that *miR-1246* was highly upregulated (3–5 fold) in patients with liver fibrosis and cirrhosis, and that it is linked to drug resistance, tumor stemness, and metastasis [68]. *miR-1246* regulates key signaling pathways, including JAK/STAT, PI3K/AKT, EMT, and TGF-β, suggesting it as a potential target in fibrosis and cirrhosis [69]. These pathways align with DCLK1 activity, implicating *miR-1246* in pro-inflammatory, stemness, and pro-tumorigenic processes, making it a candidate for further diagnostic and therapeutic investigation. Our findings highlight the importance of an unbiased miRNA discovery platform using sera from patients with chronic liver disease to identify novel targets and pathways, aiding in the development of diagnostic and therapeutic agents.

Given DCLK1’s role in inflammatory liver diseases, tumorigenesis, and metastasis [27,28], we investigated stage-specific miRNA expression in patients with high (40–60 ng/mL) and low (2–20 ng/mL) DCLK1 levels. We identified three miRNAs that changed in patients with elevated DCLK1, as follows: *miR-1246* and *miR-184* increased, while *miR-206* decreased. Patients with high DCLK1 had a fourfold increase in *miR-1246* with fibrosis but not HCC, suggesting that high *miR-1246* levels contribute to inflammation-driven tumorigenesis [70]. DCLK1-mediated *miR-1246* increase likely promotes liver disease progression (Figure 5A). This is supported by the evidence of *miR-1246* expression in pancreatic cancer stem cells, where DCLK1 also plays a role [71]. *miR-206*, a known tumor suppressor [38], was downregulated when DCLK1 was high. This suggests that high DCLK1 reduces *miR-206*, releasing signals that would otherwise prevent progression from cirrhosis to HCC. *miR-206* has been shown to alleviate NAFLD symptoms by reducing lipid accumulation and tissue damage [72]. Thus, DCLK1-mediated suppression of *miR-206* may contribute to hepatic carcinogenesis by increasing inflammation and tissue damage (Figure 5B). *miR-184*, described as both an oncogene and tumor suppressor [73], was upregulated in a DCLK1-dependent manner during the progression from cirrhosis to HCC. In HCC, *miR-184* promotes cell proliferation, migration, and invasion, and reduces apoptosis, supporting its oncogenic role [74,75,76]. Consistent with an oncogenic role, we found *miR-184* upregulated in a DCLK1-dependent manner in the progression from cirrhosis to HCC (Figure 5C). These results support the notion of a progressive pathway from fibrosis to HCC that may be modulated by factors like DCLK1 acting in multiple signaling pathways.

In summary, our study highlights the need to explore DCLK1’s role in distinguishing between pre-neoplastic liver disease and HCC amid chronic inflammation. Our findings indicate that relying solely on single protein biomarkers is insufficient for accurately distinguishing between fibrosis, cirrhosis, and HCC. The involvement of miRNAs in key processes such as stemness, EMT, drug resistance, and metastasis underscores the value of comprehensive, high-throughput approaches based on DCLK1 expression. Identifying novel early markers of fibrosis may enable targeted interventions to prevent progression to cirrhosis and HCC. Combining proteins and miRNAs as biomarker signatures represents a significant advancement in disease diagnostics, progression, and monitoring, aligning with personalized medicine. These findings are especially relevant for the veteran population, where there is a high prevalence of viral hepatitis, NASH, fatty liver disease, and alcohol-related cirrhosis.

## 4. Materials and Methods

### 4.1. Patients

The study was conducted according to the guidelines of the Declaration of Helsinki and approved by the Institutional Review Board of the University of Oklahoma Health Sciences Center; Protocol # 7153 approved 21 September 2016. Informed consent was obtained from all subjects involved in the study. Patients were included for this study if they had known hepatic fibrosis, cirrhosis or HCC of any etiology based on laboratory, radiologic or histologic features. The exclusion criteria included age less than 18 years, pregnancy or any other known or suspected cancers. A total of 102 samples were obtained from patients at the Oklahoma City VA Medical Center and of these patients 44 had fibrosis, 40 cirrhosis, and 18 HCC. An additional 168 samples were purchased from DX Biosamples, LLC including 40 control patients without known liver disease, 50 patients with hepatic fibrosis, 50 with cirrhosis, and 28 patients with known HCC. All samples from VA Medical Center patients were divided into the following groups: no fibrosis/cirrhosis or HCC (normal or control), fibrosis without cirrhosis or HCC (fibrosis), cirrhosis without HCC (cirrhosis), and those with HCC regardless of prior fibrosis/cirrhosis (HCC). Patients with underlying liver diseases with fibrosis (non-cirrhotic and non-HCC) were classified based on their FIB4 scoring system, a biomarker panel comprising age, AST, platelet count, and ALT (FIB4 = (age × AST)/(platelets × ALT)) [77]. Cirrhosis patients were classified with Child–Pugh score A–C and without HCC [78,79].

Several factors can influence DCLK1 levels in liver disease, including genetic variability, diet, alcohol consumption, toxin exposure, comorbid conditions (like diabetes or other cancers) and various medications. To minimize these confounding factors, we screened patients and excluded those with potential confounders from our analysis.

### 4.2. Sample Collection and Separation

Blood specimens were collected according to standard procedures using a Vacutainer serum tube (BD) and were transferred to the lab within 1 h of collection. The blood was left undisturbed in its entirety at room temperature and allowed to clot. The blood was fractionated in its entirety by centrifuging at 2000 rpm for 10 min at 4 °C to separate the clot and the supernatant. The supernatant serum layer was collected without disturbing the clot, transferred to a fresh sterile tube, and stored at −85 °C until use.

### 4.3. Enzyme-Linked Immunosorbent Assays (ELISA)

Serum DCLK1, TGF-β, ACE2, and AFP-L3 protein levels were quantified in patients’ plasma samples by sandwich ELISA methodology using commercially available ELISA kits. ELISA kits for DCLK1 (USCN Life Science, Inc., Wuhan, China), TGF-β, ACE 2 (both from Abcam, Cambridge, UK), and AFP-L3 (MyBioSource, San Diego, CA, USA) were used according to manufactures’ instructions. Different concentrations (0–10 ng/mL) of purified proteins of interest (as mentioned above) were used to create each respective standard curve. Serum samples were diluted 1:4 and 1:10 with PBS. The diluted serum samples or purified proteins were added, and the value of OD 450 nm was measured using a microplate reader according to the manufacture’s procedure. The concentrations of DCLK1 and other proteins in plasma samples were then determined based on the standard curve constructed using the purified protein via the SOFTMAX PRO software (v7.0) (Molecular Devices Corp., San Jose, CA, USA). For each assay, all patient samples were measured in three independent runs across three separate dates as well as being measured in triplicate wells for each run. Using this repetitive data, intra-plate, and inter-plate coefficients of variation (CV) were calculated (using the formula SD/m, where SD = standard deviation and m = mean) for each assay to assess their reproducibility. Samples with readings greater than the highest standard for any respective ELISA test were diluted appropriately and the assay repeated.

### 4.4. RNA Isolation and miRNA seqRNA

Total miRNA was extracted from serum using the miRCURY^TM^ RNA isolation kit–biofluids (Exiqon, Vedbaek, Denmark) according to the manufacturer’s instructions. Next-generation sequencing was performed by Qiagen miRNA sequencing services (Qiagen, Germantown, MD, USA). The expression of each miRNA was derived from the maximum of the average group reads per kilobase per million (RPKM) values between the two named conditions, i.e., liver disease versus normal control. The analysis of the data was performed with Qiagen CLC Genomics Workbench software (v22) or Python (v3).

### 4.5. Statistical Analysis

Protein levels among groups were compared using ANOVA. If the overall test was significant, pairwise comparisons were conducted with Bonferroni method for multiple comparisons. If data were not normally distributed, Kruskal–Wallis’s test was used for group comparison. If the overall Kruskal–Wallis’s test was significant, pairwise comparisons were conducted using the Dwass, Steel, Critchlow–Fligner (DSCF) multiple comparison. SAS software (v9.4) was used for these analyses. For miRNA analysis (performed by Qiagen), differential gene expression analyses for pairwise group comparisons were performed using the negative binomial generalized linear models for count data. We used the Wald test [80] to generate *p* values, which were corrected for multiple testing by the Benjamini and Hochberg method to control the false discovery rate (FDR). It should be noted that the fold change in the defined order between the named pair are calculated from the generalized linear model (GLM), which corrects for differences in library size between the samples and the effects of confounding factors. It is therefore not possible to derive these fold changes from the original counts by simple algebraic calculations. Results were summarized in Log_2_ fold changes and multiple-testing-corrected *p* values (labeled as FDR *p* values), focusing on those with an FDR *p* value < 0.05 and a log_2_ fold change of >1 or <−1. Principal component analysis (PCA) using 500 genes with the highest variance across samples was conducted. Hierarchal clustering using 35 genes with the highest variance across samples was also conducted. In both analyses, a variance stabilizing transformation on the raw count matrix was used.

## Figures and Tables

**Figure 1 ijms-25-06481-f001:**
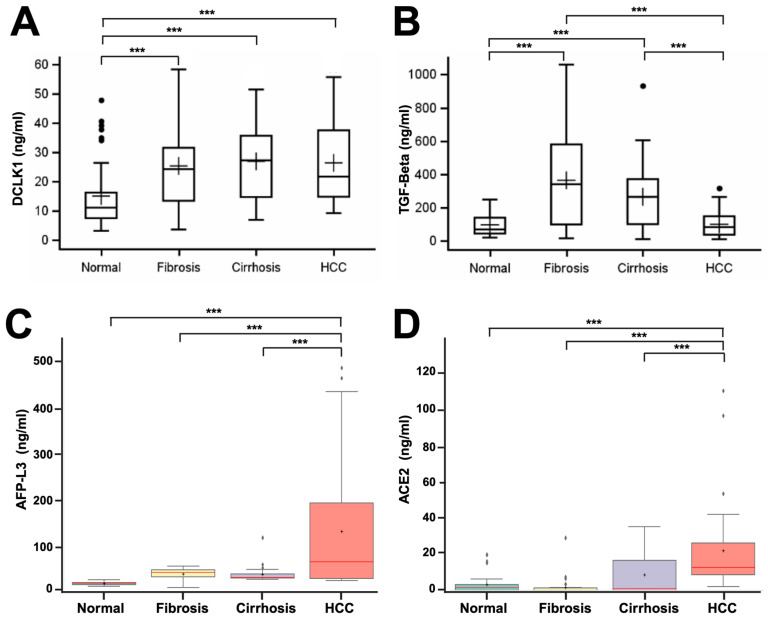
Differential blood sera protein expression of key factors in liver disease progression. Boxplots of protein levels from ELISAs performed on 168 serum samples, including 40 normal, 50 fibrosis, 50 cirrhosis, and 28 HCC patients to determine relative protein levels in these samples. (**A**) DCLK1 serum levels in patients show elevated fibrosis, cirrhosis, and HCC levels. (**B**) TGF-β show increased levels in fibrosis and cirrhosis compared with normal liver, but normal levels in HCC sera. (**C**,**D**) AFP-L3 and ACE2 protein levels were significantly elevated in HCC only. Dots represent outliers. ‘***’—*p* value < 0.01 compared with normal, ‘+’—mean.

**Figure 2 ijms-25-06481-f002:**
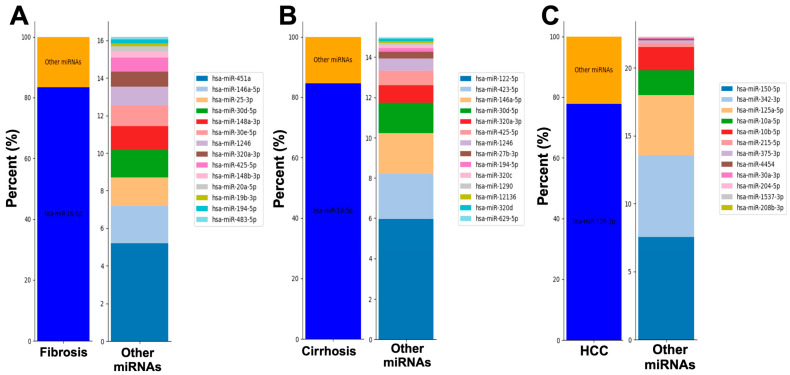
Top 15 DE miRNAs present in sera from patients with fibrosis, cirrhosis, and HCC. For each condition, the first bar shows the most abundant DE miRNA and the remainder percentages of other miRNAs. The second bar indicates the relative percentages of the next 14 most abundant DE miRNAs. (**A**) Profile of miRNAs in fibrosis patient’s sera compared with normal liver patient’s sera. (**B**) Similar DE miRNA profile of cirrhotic patient’s sera compared with normal liver. (**C**) The DE miRNA profile of the HCC patient’s sera was distinct from that of fibrotic and cirrhotic patient sera.

**Figure 3 ijms-25-06481-f003:**
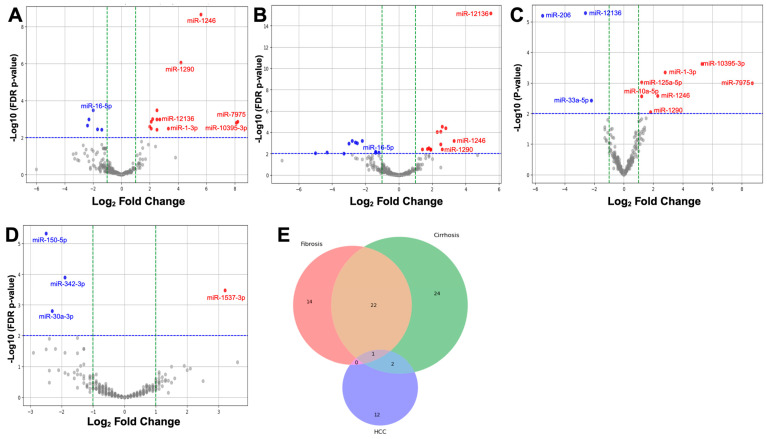
Differentially expressed (DE) miRNAs in fibrosis, cirrhosis, and HCC patient sera. (**A**–**C**) Volcano plots of indicated patient sera compared with normal liver patient sera using the false detection rate (FDR) *p* value threshold of <0.01. The log2FC indicates the mean expression level for each miRNA (dot). Red dots represent up-regulated and green represents down-regulated miRNAs. The blue line presents the significance threshold and green lines the expression cutoffs. (**A**) Fibrosis miRNAs in sera from fibrosis patients showing upregulation or downregulation relative to normal liver patient sera. Some miRNAs have been labeled for comparison. (**B**) Similar plot to (**A**) but with the cirrhosis patient sera compared with normal. (**C**) Comparison of fibrosis patient sera miRNAs versus cirrhosis patient sera miRNAs. (**D**) HCC sera showed different DE miRNAs than those seen in fibrosis or cirrhosis. (**E**) Venn diagram of DE miRNA clustering into unique fibrosis, cirrhosis, and HCC as well as each overlapping grouping.

**Figure 4 ijms-25-06481-f004:**
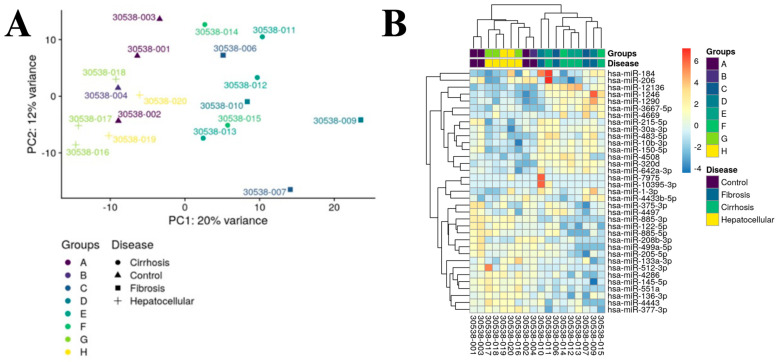
DE miRNAs vary based on DCLK1 expression. (**A**) Principal component analysis (PCA) of patient serum grouped by DCLK1 expression level. A variance stabilizing transformation was performed on the raw count matrix and 500 miRNAs with the highest variance were used to plot the PCA. The variance was calculated agnostically to the pre-defined groups. (**B**) Heatmap of hierarchal cluster analysis.

**Figure 5 ijms-25-06481-f005:**
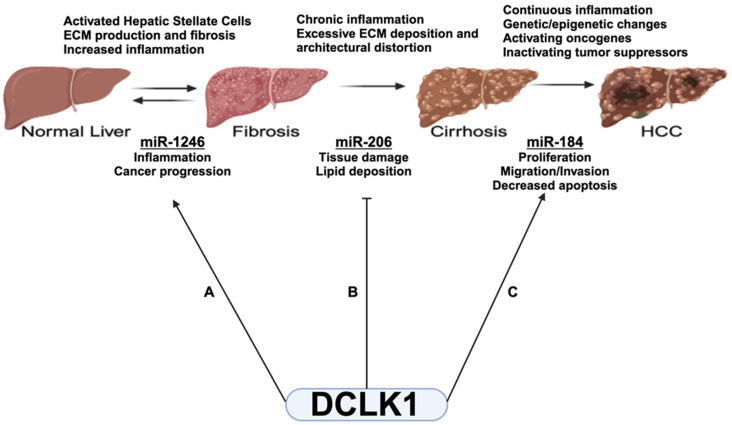
Diagram of DCLK1 and DCLK1-modulated miRNAs in liver disease progression. (**A**) Disease progression from normal liver to fibrotic liver by elevation of *miR-1246* expression, (**B**) Inhibition of *miR-206* promotes progression to cirrhosis, and (**C**) elevated *miR-184* promotes HCC. Created with BioRender.com (accessed on 28 May 2024).

**Table 1 ijms-25-06481-t001:** DE miRNAs in fibrosis progression (ranked by fibrosis expression).

miRNA	Expression (Fold Change (FC)) vs. Control						Fibrosis vs. Cirrhosis	
	Fibrosis		Cirrhosis		HCC			
	log_2_ FC	FDR *p* Value	log_2_ FC	FDR *p* Value	log_2_ FC	FDR *p* Value	log_2_ FC	FDR *p* Value
hsa-miR-7975	8.2	0.0014	−0.4	0.9741	2.3	0.6744	8.7	0.0971
hsa-miR-10395-3p	8.1	0.0016	2.8	0.5049	−0.5	0.9483	5.3	0.0454
hsa-miR-3667-5p	5.6	3.85 × 10^−5^	4.4	0.0010	3.1	0.0998	1.3	0.6963
hsa-miR-1246	5.6	2.12 × 10^−9^	3.3	0.0006	0.0	0.9892	2.3	0.1708
hsa-miR-1290	4.2	8.54 × 10^−7^	2.6	0.0040	0.5	0.8371	1.8	0.3955
hsa-miR-1-3p	3.3	0.0032	0.6	0.7926	1.0	0.5772	2.8	0.0654
hsa-miR-12136	2.7	0.0010	5.5	6.56 × 10^−19^	0.4	0.8482	−2.6	0.0018
hsa-miR-122-3p	2.7	0.0448	1.1	0.5664	1.4	0.4709	1.8	0.5825
hsa-miR-320d	2.5	0.0003	2.5	8.09 × 10^−5^	0.6	0.6744	0.1	0.9729
hsa-miR-27a-5p	2.5	0.0448	2.2	0.0293	2.0	0.1304	0.4	0.9447
hsa-miR-3679-5p	2.5	0.0010	1.8	0.0147	0.8	0.5216	0.8	0.7621
hsa-miR-642a-3p	2.5	0.0036	1.1	0.2885	−0.8	0.6105	1.5	0.5027
hsa-miR-483-5p	2.4	0.0306	1.3	0.2812	−0.8	0.6684	1.3	0.6207
hsa-miR-125a-3p	2.3	0.0343	1.0	0.4715	0.9	0.5302	1.4	0.5825
hsa-miR-3960	2.2	0.0010	2.3	9.47 × 10^−5^	−0.3	0.8637	0.0	0.9803
hsa-miR-627-5p	2.1	0.0032	2.6	2.78 × 10^−5^	0.1	0.9462	−0.4	0.8832
hsa-miR-320c	2.1	0.0014	1.8	0.0029	0.1	0.9554	0.4	0.8832
hsa-miR-320b	2.0	0.0025	1.6	0.0125	0.0	0.9938	0.5	0.8255
hsa-miR-4429	1.9	0.0238	1.9	0.0043	0.1	0.9676	0.1	0.9729
hsa-miR-629-5p	1.5	0.0355	1.7	0.0037	0.7	0.4360	−0.1	0.9729
hsa-miR-320a-3p	1.4	0.0162	1.4	0.0040	−0.4	0.6744	0.1	0.9729
hsa-miR-30e-5p	−1.1	0.0379	−0.6	0.2654	0.0	0.9880	−0.3	0.8255
hsa-miR-148b-3p	−1.1	0.0405	−0.9	0.0991	0.0	0.9720	−0.1	0.9571
hsa-miR-20a-5p	−1.2	0.0379	−0.4	0.6584	−0.1	0.9148	−0.7	0.6176
hsa-miR-425-5p	−1.2	0.0377	−1.1	0.0283	−0.3	0.7935	0.0	0.9818
hsa-miR-146a-5p	−1.2	0.0379	−1.4	0.0063	−0.3	0.7528	0.2	0.9447
hsa-miR-19b-3p	−1.3	0.0386	−0.3	0.7926	0.4	0.6760	−0.8	0.5825
hsa-miR-148a-3p	−1.3	0.0383	−0.9	0.1440	0.2	0.8482	−0.2	0.9447
hsa-miR-25-3p	−1.4	0.0238	−0.8	0.2350	−0.6	0.5162	−0.4	0.8255
hsa-miR-30d-5p	−1.4	0.0037	−1.2	0.0075	−0.4	0.5640	0.0	0.9803
hsa-miR-451a	−1.7	0.0036	−0.8	0.2812	−0.8	0.3722	−0.8	0.6176
hsa-miR-16-5p	−2.0	0.0003	−1.4	0.0079	−1.0	0.1979	−0.4	0.8255
hsa-miR-499a-5p	−2.5	0.0434	−3.0	0.0011	−1.5	0.1811	0.7	0.8889
hsa-miR-16-5p	−2.0	0.0003	−1.4	0.0079	−1.0	0.1979	−0.4	0.8255
hsa-miR-200a-3p	−2.1	0.0448	−2.6	0.0009	−1.4	0.1811	0.7	0.8255
hsa-miR-194-5p	−2.3	0.0010	−2.2	0.0006	−1.0	0.3269	0.1	0.9803

**Table 2 ijms-25-06481-t002:** DE miRNAs in cirrhosis to HCC progression (ranked by cirrhosis expression).

miRNA	Expression (Fold Change (FC)) vs. Control						Cirrhosis vs. HCC Fibrosis	
	Fibrosis		Cirrhosis		HCC			
	log_2_ FC	FDR *p* Value	log_2_ FC	FDR *p* Value	log_2_ FC	FDR *p* Value	log_2_ FC	FDR *p* Value
hsa-miR-184	3.8	0.1178	4.7	0.0140	1.4	0.7145	3.2	0.0896
hsa-miR-1297	1.4	0.3037	2.8	4.01 × 10^−5^	0.2	0.9127	2.6	7.20 × 10^−5^
hsa-miR-4488	1.7	0.1471	2.5	0.0013	−0.4	0.8482	2.8	8.46 × 10^−5^
hsa-miR-206	−3.1	0.1620	2.5	0.1879	−2.4	0.3269	4.8	0.0007
hsa-miR-4508	1.2	0.4321	2.0	0.0319	−1.0	0.5143	2.9	0.0002
hsa-miR-576-3p	1.2	0.2533	1.9	0.0040	0.9	0.3926	0.9	0.1556
hsa-miR-3605-5p	1.3	0.3328	1.8	0.0195	−0.5	0.8062	2.3	0.0026
hsa-miR-4516	1.2	0.2508	1.6	0.0261	−0.3	0.8421	1.9	0.0027
hsa-miR-651-5p	1.3	0.0742	1.3	0.0318	0.6	0.6105	0.8	0.2316
hsa-miR-423-5p	1.1	0.0874	1.2	0.0293	0.1	0.9608	1.1	0.0236
hsa-miR-27b-3p	−0.6	0.1568	−1.2	0.0147	−0.6	0.0978	−0.6	0.0852
hsa-miR-598-3p	−0.9	0.5319	−1.6	0.0286	−0.6	0.5640	−1.0	0.2140
hsa-miR-132-3p	−0.6	0.7290	−1.7	0.0261	−0.8	0.4654	−0.9	0.2907
hsa-miR-141-3p	−1.3	0.3132	−1.7	0.0319	−1.5	0.1314	−0.2	0.8700
hsa-miR-34a-5p	−1.2	0.3394	−1.8	0.0195	−0.3	0.8868	−1.6	0.0403
hsa-miR-483-3p	−0.6	0.7307	−1.9	0.0273	−1.6	0.0273	−0.3	0.2842
hsa-miR-205-5p	−1.3	0.3095	−2.5	0.0010	−1.3	0.2030	−1.2	0.1555
hsa-miR-429	−0.7	0.7321	−2.5	0.0293	−0.5	0.8208	−2.1	0.0751
hsa-miR-885-5p	−1.1	0.6561	−2.6	0.0261	−0.4	0.8963	−2.3	0.0457
hsa-miR-122-5p	−1.8	0.0773	−2.8	0.0006	−0.9	0.5216	−1.9	0.0251
hsa-miR-200b-3p	−0.9	0.6838	−2.8	0.0122	−0.6	0.7145	−2.3	0.0494
hsa-miR-296-5p	−0.5	0.7767	−3.3	0.0097	−0.6	0.6882	−2.7	0.0403
hsa-miR-208b-3p	−3.3	0.1386	−4.3	0.0076	−2.9	0.0361	−1.5	0.4826
hsa-miR-1299	−3.0	0.1775	−5.0	0.0093	−2.4	0.1320	−2.6	0.2324
hsa-miR-885-3p	−3.4	0.0773	−7.0	0.0421	−0.3	0.8679	−6.6	0.0443

**Table 3 ijms-25-06481-t003:** DE miRNAs in progression towards HCC (ranked by HCC expression).

miRNA	Expression (Fold Change (FC)) vs. Control		Fibrosis vs. HCC		Cirrhosis vs. HCC Fibrosis	
	HCC					
	log_2_ FC	FDR *p* Value	log_2_ FC	FDR *p* Value	log_2_ FC	FDR *p* Value
hsa-miR-132-5p	3.6	0.0280	--	--	−2.9	0.0618
hsa-miR-1537-3p	3.2	0.0003	−3.4	0.0262	−2.9	0.0025
hsa-miR-126-3p	−1.3	0.0267	1.9	1.20 × 10^−5^	1.2	0.0157
hsa-miR-125a-5p	−1.3	0.0267	2.2	1.69 × 10^−7^	1.0	0.0333
hsa-miR-10b-5p	−1.5	0.0372	2.0	0.0003	0.9	0.1599
hsa-miR-10a-5p	−1.5	0.0120	1.8	0.0002	0.6	0.2907
hsa-miR-342-3p	−1.9	0.0001	2.2	4.38 × 10^−7^	1.3	0.0093
hsa-miR-4454	−1.9	0.0361	0.0	0.9991	1.3	0.1332
hsa-miR-204-5p	−2.2	0.0267	2.8	0.0008	2.0	0.0242
hsa-miR-30a-3p	−2.3	0.0016	3.3	1.77 × 10^−7^	2.3	0.0005
hsa-miR-215-5p	−2.4	0.0125	2.6	0.0014	1.3	0.1700
hsa-miR-150-5p	−2.5	4.68 × 10^−6^	3.4	9.96 × 10^−12^	2.3	3.88 × 10^−6^
hsa-miR-375-3p	−2.5	0.0273	0.0	0.9868	0.3	0.8306

**Table 4 ijms-25-06481-t004:** DE miRNAs based on DCLK1 expression levels (fold change between DCLK1 high vs. low).

miRNA	Fibrosis			Cirrhosis			HCC		
	Log_2_ Fold Change	Fold Change	FDR *p* Value	Log_2_ Fold Change	Fold Change	FDR *p* Value	Log_2_ Fold Change	Fold Change	FDR *p* Value
hsa-miR-184	−4.6	−25	N.D.	4.2	18.4	0.3693	−6.9	−121.7	0.0305
hsa-miR-1246	−3.8	−13.6	0.0151	−1.3	−2.4	0.9102	−0.9	−1.8	0.9983
hsa-miR-206	0.1	1.1	0.9431	5.7	50.7	0.028	−1.8	−3.4	0.9983

N.D.—not determined.

## Data Availability

The raw data supporting the conclusions of this article will be made available by the authors on request.

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
