# Peer review of "From Inflammation to Oncogenesis: Tracing Serum DCLK1 and miRNA Signatures in Chronic Liver Diseases"

_ijms, 2024, doi:10.3390/ijms25126481_

Round 1

Reviewer 1 Report

Comments and Suggestions for Authors

The study identified elevated serum DCLK1 levels and miRNAs in fibrosis, cirrhosis, and HCC patients, which may be served as biomarkers for tracking chronic liver diseases. This work is innovative and interesting. I have some comments as follows.

1.       In the introduction, the role of DCLK1 and the previous study have been elaborated. However, the role of miRNAs in the chronic liver diseases has not been described.

2.       Is the level of miRNA in the patient's serum obtained through sequencing or detection in the results? If it is only the result of sequencing without validation, the level of miRNA in the patient's serum should be detected.

3.       The article mentions that DCLK1 regulates miRNA. How to determine the upstream and downstream relationship between miRNA and DCLK1? Generally speaking, miRNA regulates the expression of proteins.

4.       “Incorporating both proteins and miRNAs as biomarker signatures constitutes a pivotal advancement in the realm of disease diagnostics, progression, and monitoring, thus aligning with the new paradigm of personalized medicine.” As mentioned in the discussion, incorporating both proteins and miRNAs as biomarker is better than the single one. However, there is no data in the article for both joint and single detection. Does it have higher sensitivity and specificity than single detection.

5.       FDR, appearing for the first time in the results without specifying the full name.

Author Response

REVIEWER #1

The study identified elevated serum DCLK1 levels and miRNAs in fibrosis, cirrhosis, and HCC patients, which may be served as biomarkers for tracking chronic liver diseases. This work is innovative and interesting. I have some comments as follows.

  1. In the introduction, the role of DCLK1 and the previous study have been elaborated. However, the role of miRNAs in the chronic liver diseases has not been described.

Response: We thank the reviewer for pointing this out.  In the revised manuscript we have added an additional paragraph that briefly reviews the current landscape of miRNAs in liver disease.

  1. Is the level of miRNA in the patient's serum obtained through sequencing or detection in the results? If it is only the result of sequencing without validation, the level of miRNA in the patient's serum should be detected.

Response: We used Next-Generation Sequencing as the technical method for detection of miRNA and the expression of each miRNA was obtained from the Max group mean i.e. the maximum of the average group RPKM values between the two named tissue types (Conditions). This was added to the methods. We will provide this data upon request or upload to a data repository.

  1. The article mentions that DCLK1 regulates miRNA. How to determine the upstream and downstream relationship between miRNA and DCLK1? Generally speaking, miRNA regulates the expression of proteins.

Response: We apologize and thank the reviewer for illustrating this oversight.  We have added content in the introduction section highlighting reports demonstrating that DCLK1 is regulated by specific miRNAs, however DCLK1 protein itself can also influences the expression of other downstream miRNAs thus providing for a unique network of regulation.

  1. “Incorporating both proteins and miRNAs as biomarker signatures constitutes a pivotal advancement in the realm of disease diagnostics, progression, and monitoring, thus aligning with the new paradigm of personalized medicine.” As mentioned in the discussion, incorporating both proteins and miRNAs as biomarker is better than the single one. However, there is no data in the article for both joint and single detection. Does it have higher sensitivity and specificity than single detection.

Response: We fully understand the reviewers concerns, however, the fundamental goal of this study was designed to identify putative miRNAs that could contribute to phenotypic differences that occur during the progression of inflammatory liver diseases based on our earlier findings demonstrating that serum DCLK1 is elevated in patients during the progression of liver disease towards HCC, as such the study was not designed as a biomarker identification and validation tool. We are fully aware however that further mechanistic studies evaluating the role of miRNAs in the context of elevated DCLK1 may provide tools to identify novel biomarker candidates and signatures.  However, this is beyond the scope of the current study.

  1. FDR, appearing for the first time in the results without specifying the full name.

Response: Thank you very much and we have corrected these errors in the revised manuscript.

Reviewer 2 Report

Comments and Suggestions for Authors

The author detected serum DCLK1 and miRNA for indicating progress of liver inflammation and cancer. This work found serum DCLK1 and some miRNA can serve as biomarker for liver diseases. Generally, the manuscript is well prepared and presented, and it sounds interesting for academic. But there are some improvements to be made before it can be officially published.

1.         In the Introduction part, the role of miRNA in liver diseases should also be described;

2.         The author should discuss of other three proteins’ results in the discussion part.

3.         Some format errors should be omitted, such there is double space for “Further,    DCLK1 promotes hepatocyte clonogenicity and”, “following chronic inflammatory hepatic injury.    In this report, in serum samples from”, “gression in patients at specific stages in the chronic liver disease spectrum.    These miRNA”.

4.         Please double check the format of reference, such as page NO is lacking for Refs 6, 17, 20 and 36; Journal name is wrong for Ref 24; Some journal names are full name, while some journal are abbreviated.

Author Response

REVIEWER #2

The author detected serum DCLK1 and miRNA for indicating progress of liver inflammation and cancer. This work found serum DCLK1, and some miRNA can serve as biomarker for liver diseases. Generally, the manuscript is well prepared and presented, and it sounds interesting for academic. But there are some improvements to be made before it can be officially published.

  1. In the Introduction part, the role of miRNA in liver diseases should also be described.

Response: We thank the reviewer for pointing this out. We have added a paragraph that briefly reviews the current state of miRNAs in liver disease in the introduction.

  1. The author should discuss of other three proteins’ results in the discussion part.

Response: In the revised manuscript we have added additional discussion related to TGF-b, AFP-L3, and ACE2 based on our results.

  1. Some format errors should be omitted, such there is double space for “Further, DCLK1 promotes hepatocyte clonogenicity and”, “following chronic inflammatory hepatic injury. In this report, in serum samples from”, “gression in patients at specific stages in the chronic liver disease spectrum. These miRNA”.

Response: We have corrected these typographical errors.

  1. Please double check the format of reference, such as page NO is lacking for Refs 6, 17, 20 and 36; Journal name is wrong for Ref 24; Some journal names are full name, while some journal are abbreviated.

Response: All references were downloaded from publishers’ websites. Where we could identify page numbers, we have updated the entry. We corrected the Journal name on the Ref 24 (original version).

Reviewer 3 Report

Comments and Suggestions for Authors

The manuscript "From Inflammation to Oncogenesis: Tracing Serum DCLK1 and miRNA Signatures in Chronic Liver Diseases" aims to investigate the role of DCLK1 and miRNAs in chronic liver diseases' progression. Elevated DCLK1 levels were observed in fibrosis, cirrhosis, and HCC patients. DE miRNAs correlated with DCLK1 expression, indicating potential biomarkers for disease progression and therapeutic targets. Although the paper presents a certain scientific interest, there are concerns regarding the data's validity and overall results. Here are some important comments:

- Kindly provide precise numbers or ranges for the intra-plate and inter-plate coefficient of variation (CV) to showcase the reproducibility of the test.
- Kindly furnish details regarding the specificity and sensitivity of the ELISA kit, together with measures to mitigate interferences.
- When selecting patients, it is important to consider their demographic traits to ensure a more accurate representation of the cohort.
- Please improve comprehension of the regulatory relationships involved in the development of liver diseases.
- Kindly conduct a thorough examination of the cause-and-effect relationship between DCLK1 and the miRNAs that have been identified.
- Please provide an analysis of the factors that contribute to variations in miRNA expression during different phases of an illness.
- Please provide further explanation regarding the absence of substantial variation in DCLK1 levels among different stages of liver disease.
- Please provide a discussion on the potential confounding factors that may influence the levels of DCLK1.
- Kindly elucidate the diagnostic efficacy of TGF-β in differentiating hepatocellular carcinoma (HCC) from pre-neoplastic stages.
- Kindly offer a detailed explanation of the underlying mechanisms that determine the patterns of its expression throughout different phases of the disease.
- Kindly verify the reported rise in AFP-L3 and ACE2 concentrations.
- Kindly analyze the diagnostic capacity of AFP-L3 in differentiating hepatocellular carcinoma (HCC) from cirrhosis.
- Please elucidate the discriminatory capacity of ACE2 in distinguishing between fibrosis/cirrhosis and HCC.
- Please offer a detailed explanation of miRNA profiles in liver disorders.
- Please provide a rationale for selecting statistical tests in miRNA analysis, taking into account information about the distribution of the data and making explicit modifications for multiple testing.
- Kindly include the specific type of blood collection tubes to guarantee uniformity in subsequent studies.

Comments on the Quality of English Language

Some minor editing is needed to improve the English language used in the text. 

Author Response

REVIEWER #3

The manuscript "From Inflammation to Oncogenesis: Tracing Serum DCLK1 and miRNA Signatures in Chronic Liver Diseases" aims to investigate the role of DCLK1 and miRNAs in chronic liver diseases' progression. Elevated DCLK1 levels were observed in fibrosis, cirrhosis, and HCC patients. DE miRNAs correlated with DCLK1 expression, indicating potential biomarkers for disease progression and therapeutic targets. Although the paper presents a certain scientific interest, there are concerns regarding the data's validity and overall results. Here are some important comments:

  1. Kindly provide precise numbers or ranges for the intra-plate and inter-plate coefficient of variation (CV) to showcase the reproducibility of the test.

Response: We understand the reviewers concerns, however, this study was designed to identify putative miRNAs that could contribute to phenotypic differences that occur during the progression of inflammatory liver diseases, as such the study was not designed as a biomarker identification and validation tool. We fully understand though following further mechanistic evaluation that potential biomarker candidates may be identified, however, these studies were beyond the scope of the funding mechanism used for this study.

  1. Kindly furnish details regarding the specificity and sensitivity of the ELISA kit, together with measures to mitigate interferences.

Response: All kits were commercially available. As such per the manual for DCLK1 kit, the minimum detectable dose (MDD) is typically less than 0.054 ng/mL and the Lower Limit of Detection (LLD), or sensitivity is defined as the lowest protein concentration that could be differentiated from zero.  This value was determined by adding two standard deviations to the mean optical value of twenty zero replicates and calculating the corresponding concentration.  This assay has high sensitivity for DCLK1 and excellent specificity. For the ACE2 kit, it has high sensitivity that was determined by calculating the mean of zero standard replicates and adding two standard deviations then extrapolating the corresponding concentration and excellent specificity. The AFPL3 kit has high sensitivity in this assay which is 1.0 ng/mL and excellent specificity. Lastly, the MDD for Human TGF-Beta was determined to be 18 pg/ml.

  1. When selecting patients, it is important to consider their demographic traits to ensure a more accurate representation of the cohort.

Response: For the normal (control) samples, we purchase from DX Biosamples that represent 40 Caucasian without liver fibrosis, cirrhosis, or HCC. Additional samples were also purchased, however, for all DX Biosamples demographic data was not provided. Thus, we were limited to our VA Medical Center patients for demographic data. While we can provide this information it is of limited value for a biomarker study, thus we focused only on identify proteins/miRNAs that were differentially expressed between the different disease states. For our samples, VA patient sample demographics were:

Cirrhosis

29 White Males   1 White Female

2 Hispanic Male

4 Black Males

3 American Indian

2 Unknown

Fibrosis

27 White Males   3 White Females

2 Hispanic Males

5 Black Males

1 Asian Male

2 Pacific Islander

1 American Indian

HCC

17 White Males

1 Black Male

As the represents less than half of the subjects in this study, we did not include demographic data or whether it functions as a biomarker in this study. However, the criteria for each disease state were consistent, therefor we focused on identifying differentially expressed proteins and miRNAs as candidates for further study.

  1. Please improve comprehension of the regulatory relationships involved in the development of liver diseases.

Response: We added a paragraph detailing the development of liver disease.

  1. Kindly conduct a thorough examination of the cause-and-effect relationship between DCLK1 and the miRNAs that have been identified.

Response: We attempted to provide a thorough examination of the cause-and-effect relationship between DCLK1 and the miRNAs, however, we are limited to current understanding for the miRNAs. What we provide here is a first link between these miRNAs and DCLK1, which provides for further investigation.

  1. Please provide an analysis of the factors that contribute to variations in miRNA expression during different phases of an illness.

Response: We thank the review for this insight, and we have added additional text further defining our focus on miRNA expression in the different phases of liver disease.

  1. Please provide further explanation regarding the absence of substantial variation in DCLK1 levels among different stages of liver disease.

Response: We thank the reviewer and have revised to the manuscript to provide further explanation as to the absence of substantial variation in DCLK1 levels among the different stages of liver disease and the potential role changes that may be involved.

  1. Please provide a discussion on the potential confounding factors that may influence the levels of DCLK1.

Response: We agree that confounding factors may influence the levels of DCLK1 and other factors. However, during patient recruitment we endeavored to remove samples with confounding factors, specifically those with comorbid conditions or excessive alcohol consumption where possible to identify. We have added a concise statement in the Methods to this effect so as not to clutter the Discussion.

  1. Kindly elucidate the diagnostic efficacy of TGF-β in differentiating hepatocellular carcinoma (HCC) from pre-neoplastic stages.

Response: We have added content on the efficacy of TGF-β in differentiating hepatocellular carcinoma (HCC) from pre-neoplastic stages to the discussion.

  1. Kindly offer a detailed explanation of the underlying mechanisms that determine the patterns of its expression throughout different phases of the disease.

Response: We have added content on the expression of TGF-β in the preneoplastic stages of HCC versus HCC to the discussion.

  1. Kindly verify the reported rise in AFP-L3 and ACE2 concentrations.

Response: We apologize for this omission and have included these values in the Results.

  1. Kindly analyze the diagnostic capacity of AFP-L3 in differentiating hepatocellular carcinoma (HCC) from cirrhosis.

Response: We have added discussion on the diagnostic capacity of AFP-L3 in distinguishing cirrhosis from HCC.

  1. Please elucidate the discriminatory capacity of ACE2 in distinguishing between fibrosis/cirrhosis and HCC.

Response: We observed that ACE2 is upregulated in HCC relative to fibrosis/cirrhosis as others have. Our study was not able to characterize the discriminatory capacity of ACE2 as our focus was on distinguishing between the early phases of liver disease prior to HCC.

  1. Please offer a detailed explanation of miRNA profiles in liver disorders.

Response: We have included in the revised manuscript a more detailed explanation of miRNA profiles in liver disease.

  1. Please provide a rationale for selecting statistical tests in miRNA analysis, taking into account information about the distribution of the data and making explicit modifications for multiple testing.

Response: We added additional methods explaining how the miRNA data was analyzed in the Methods section.

  1. Kindly include the specific type of blood collection tubes to guarantee uniformity in subsequent studies.

Response: For this study we used BD vacutainer red top serum tube 6.0mL. For reproducibility we have added this information to the Materials and Methods section.

Reviewer 4 Report

Comments and Suggestions for Authors

see attached

Comments on the Quality of English Language

Only minor spelling errors were found (e.g. Results, 2.3 2nd paragraph: "comparing" is written with capital C)

Author Response

REVIEWER #4

Only minor spelling errors were found (e.g. Results, 2.3 2nd paragraph: "comparing" is written with capital C)

Response: These have been corrected.

This is an interesting study reporting on potential biomarkers in liver fibrosis/cirrhosis and HCC. I have a few remarks.

Introduction:

  1. I suggest to remove/paste the last few sentences beginning with “Secondly, we determined the downstream…” to Results, as it belong there.

Response: We have moved the suggested section to the appropriate Results section.

Results:

  1. The sentence “These results demonstrate that in this population…” belongs into Discussion.

Response: We agree and moved this to the discussion.

  1. Explanation/background information regarding AFP, ACE2 incl. sources (19, 4,20,22): Giving explanations/background information in Results is confusing. I suggest to move them to Introduction or Discussion.

Response: We moved the background information on AFP and ACE2 to the introduction to clarify the results section.

  1. Figure 2: I suggest to add another figure showing healthy controls.

Response: We apologize for the confusion. This figure represents the top differential expressed (DE) miRNAs when compared to control samples. We have edited the figure legend to make this clearer.

Discussion:

  1. It would be great if the authors could add an additional diagram/Figure with the different hypothesized pathways from DCLK1, its pathway and the most important miRNAs to either fibrosis, cirrhosis, and development to HCC to help understand common and diverging aspects.

Response: We thank the reviewer for this suggestion, and we have added Figure 5 that covers the three miRNAs modulated by DCLK1 and their likely action in each step of liver disease progression.

Methods:

  1. First sentence can be left out.

Response: We agree and removed the superfluous sentence.

  1. How many patients were included without histologic proof of fibrosis/cirrhosis/HCC? Was histological staging conducted and was changes in e.g. DCLK/miRNAs in correspondence with histological staging?

Response: No, we did not conduct this staging as it was not part of our original protocol that was focused on solely identifying patients with each disease condition independent of DCLK1 or miRNAs status. Thus, we sought to avoid bias, however, we agree a further study involving finer staging would be informative.

  1. Did HCC profiles differ regarding grade of differentiation (G1-G4?)

Response: Again, we did not conduct this staging as it was not part of our original protocol that was focused on solely identifying patients with HCC.

  1. Did the authors investigate a certain combination of biomarkers to allow the most precise prediction of underlying liver disease or its extent/severity?

Response: We understand the reviewers concerns, however, this study was designed to identify putative miRNAs that could contribute to phenotypic differences that occur during the progression of inflammatory liver diseases, as such the study was not designed as a biomarker identification and validation tool. We fully understand though following further mechanistic evaluation that potential biomarker candidates may be identified, however, these studies were beyond the scope of the funding mechanism used for this study.

Round 2

Reviewer 1 Report

Comments and Suggestions for Authors

1. TGF-β cannot be displayed properly in the text.

Author Response

TGF-β cannot be displayed properly in the text.

Response: Thank you for identifying this issue. We have reviewed the document and identified and corrected each instance.

Reviewer 3 Report

Comments and Suggestions for Authors

The paper has been significantly improved and is now much clearer. I believe that the authors have addressed enough comments to meet the acceptance criteria for publication in IJMS. 

Author Response

The paper has been significantly improved and is now much clearer. I believe that the authors have addressed enough comments to meet the acceptance criteria for publication in IJMS. 

Response: We thank the reviewer for their insightful comments helping to improve the manuscript.

Reviewer 4 Report

Comments and Suggestions for Authors

The authors have adressed all remarks.

Comments on the Quality of English Language

Minor English editing would be useful.

Author Response

Minor English editing would be useful.

Response: Thank you for your valuable feedback. We have reviewed the manuscript and made the necessary minor English edits to improve clarity and readability. We appreciate your attention to detail and constructive suggestions.